# Separation, Characterization, and Handling of Microalgae by Dielectrophoresis

**DOI:** 10.3390/microorganisms8040540

**Published:** 2020-04-09

**Authors:** Vinzenz Abt, Fabian Gringel, Arum Han, Peter Neubauer, Mario Birkholz

**Affiliations:** 1Chair of Bioprocess Engineering, Department of Biotechnology, Technische Universität Berlin, Ackerstr. 76, ACK24, 13355 Berlin, Germany; v.abt@campus.tu-berlin.de (V.A.); peter.neubauer@tu-berlin.de (P.N.); 2IHP–Leibniz-Institut für innovative Mikroelektronik, Im Technologiepark 25, 15236 Frankfurt (Oder), Germany; fabian.gringel@gmail.com; 3Department of Electrical and Computer Engineering, Texas A&M University, College Station, TX 77843, USA; arum.han@ece.tamu.edu

**Keywords:** microalgae, dielectrophoresis, cell sorting, microfluidics

## Abstract

Microalgae biotechnology has a high potential for sustainable bioproduction of diverse high-value biomolecules. Some of the main bottlenecks in cell-based bioproduction, and more specifically in microalgae-based bioproduction, are due to insufficient methods for rapid and efficient cell characterization, which contributes to having only a few industrially established microalgal species in commercial use. Dielectrophoresis-based microfluidic devices have been long established as promising tools for label-free handling, characterization, and separation of broad ranges of cells. The technique is based on differences in dielectric properties and sizes, which results in different degrees of cell movement under an applied inhomogeneous electrical field. The method has also earned interest for separating microalgae based on their intrinsic properties, since their dielectric properties may significantly change during bioproduction, in particular for lipid-producing species. Here, we provide a comprehensive review of dielectrophoresis-based microfluidic devices that are used for handling, characterization, and separation of microalgae. Additionally, we provide a perspective on related areas of research in cell-based bioproduction that can benefit from dielectrophoresis-based microdevices. This work provides key information that will be useful for microalgae researchers to decide whether dielectrophoresis and which method is most suitable for their particular application.

## 1. Recent Advances in Microalgae Research and Microalgae-Based Bioprocess Development

Microalgae are photosynthetic microorganisms that can utilize sunlight and CO_2_ to produce diverse ranges of bioproducts, including various high-value lipids and pigments to name a few [1,2,3]. Several applications, especially sustainable biofuel production from microalgae, are considered essential in the United Nation’s (UN) Sustainable Development Goals and for the European Union’s (EU) bioeconomy strategy [4,5]. While the current generation of microalgae-based biofuels is not economically competitive yet, next-generation fuels coming from genetically modified microalgae with much higher yields and much higher quality lipids, are emerging thanks to the advances made in biotechnology and synthetic biology. An example is *Nanochloropsis gaditana*, which was recently shown to double its production rate by simply decreasing the expression of a single regulator [6]. Besides biofuel, microalgae are interesting cell factories in many other fields [7], such as the cosmetic industry, as well as for pharmaceutical applications where microalgae are sources of large-scale production of anti-inflammatory, anti-microbial, anti-viral, and anti-tumor molecules [8]. Additionally, the use of microalgae as either animal feedstock or as a nutritional source for the growing global population is promising [9] (Figure 1). For example, microalgae are an important source of polyunsaturated fatty acid (PUFA). Currently, docosahexaenoic acid (DHA) from microalgae is contained in baby food, but also DHA-producing microalgae can be directly applied as fish feed in aquaculture if effective production processes at a larger scale can be established [7,10,11]. Many commercialization efforts in this field are also ongoing. For example, recently a joint venture of Evonik and DSM was established (Veramaris™), aiming at the development of large-scale processes of DHA with the microalgae *Schizochytrium sp.* to provide 15% of the global need of DHA. In parallel, discovering previously unknown or unidentified microalgae from nature continues [12]. Ongoing recent bioprospecting efforts [13,14] show that there us still a large number of unexplored microalgae that we are not significantly aware of, leaving the possibility of further bioprospecting efforts to uncover microalgal strains with interesting bioproduction capabilities.

Despite these advances, many challenges still remain [15,16,17]. Identifying and developing strains with higher productivity, improved understanding of their behaviors under diverse cultivation conditions, and better insights into the heterogeneous population are all some of the key advances that have to be made in the upstream process of microalgae biotechnology and bioprocessing [16,18,19,20,21,22]. Yet, these development processes are often time-consuming and labor-intensive, posing as a significant bottleneck. Part of these challenges is due to the lack of rapid and efficient instruments and methods that have been limiting rapid advancements in this field [12,23].

Development of various microfluidic devices has revolutionized the area of cell analysis, separation, and cultivation in many fields of biotechnology in the past two decades due to the single-cell resolution and high-throughput capabilities of such systems [24,25,26]. The application of these powerful technologies has recently been missed in various microalgae biotechnology fields [27,28]. General applications and impacts of microfluidic devices for microalgae have been presented [28]. In this review paper, we will specifically focus on dielectrophoresis (DEP)-based characterization, manipulation, and separation techniques in microalgae biotechnology applications.

## 2. DEP Technology for Cell Population Analysis

Flow cytometry, in general, is the current gold standard for many high-throughput single-cell analysis applications [29]. Fluorescence-based methods such as fluorescence-activated cell sorting (FACS) are widely utilized to determine the characteristics of cells based on fluorescent staining of particular target molecules, such as intracellular lipids in microalgae [30,31]. Besides fluorescence, direct imaging of cells to determine their phenotypes and/or characteristics is also becoming increasingly popular thanks to the advances in imaging flow cytometry [32,33]. Although extremely powerful and versatile, several limitations exist [18]. Most of these analyses rely on labeling cells with various markers, which require sample preparation that is time-consuming and may change the natural characteristics of cells. Additionally, flow cytometry-based methods come with relatively expensive instruments, which further limits their applications.

In DEP-based cell manipulation and sorting, the movement of polarizable particles, such as cells, in an inhomogeneous electric field is dependent on their intrinsic dielectric properties in comparison to the dielectric properties of the surrounding liquid. Thus, DEP is a noninvasive and label-free cell manipulation method. In contrast to electrophoresis, where only charged particles can be manipulated, DEP enables the interaction with uncharged particles through the induction of a dipole moment *p* under an inhomogeneous electric field. The included dipole interacts with the electrical field gradient ∇*E* [34,35], resulting in the dielectrophoretic force ***F**_DEP_* that depends on the polarizability of the particle (index *p*) within the medium (*m*), where the latter can be expressed by a function of their complex dielectric constants $\varepsilon_{p}^{*}\left(\omega \right)$ and $\varepsilon_{m}^{*}\left(\omega \right)$. These functions depend on the real part of permittivity $\varepsilon^{\prime }$ and electrical conductivity σ, and are, therefore, dependent on the frequency *f* or angular frequency ω=2πf, respectively, by which the applied voltage is oscillating, since $\varepsilon^{*}=\varepsilon^{\prime }+i\sigma /\omega$. Accordingly, ***F**_DEP_* exerted upon a spherical particle in an *AC* field is given by:(1)FDEP=π4dp3·εm′Re{εp*−εm*εp*+2εm*}·∇|E→|2,  1     2   3
where *d_p_* stands for the particle diameter and *Re* indicates the real part of the complex term inside the parenthesis. From each of the three factors, a specific feature of ***F**_DEP_* can be derived:The particle diameter *d* is the single most important contributor to ***F**_DEP_*.***F**_DEP_* depends on ε and σ of both the particle and media (index *p* and *m*). Changes of angular frequency ω may either cause a positive or negative DEP force, depending on whether $Re\left(\varepsilon_{p}^{*}-\varepsilon_{m}^{*}\right)>0\;\mathrm{or}\;Re\left(\varepsilon_{p}^{*}-\varepsilon_{m}^{*}\right)<0$ holds.The local electrical field depends on the applied voltage *V* as well as the electrode design and geometry. The layout of the flow channel and the electrodes thus offer various degrees of freedom for optimizing the DEP force.

Equation (1) represents the simplest approach for modeling the DEP effect on a homogeneous particle. A more advanced approach is utilizing a core-shell model (instead of considering the particle as a homogeneous spherical object), which is valuable if analyzing individual cellular components such as cell membranes and various intracellular components. Other extensions of the basic formula given above include the consideration of the asphericity of cells and their modeling cells as ellipsoids [36]. More details regarding complex calculations on the interaction of elliptic particles within inhomogeneous electric fields and simulation thereof are given in publications focused on DEP theory [37,38] (Figure 2).

Because of the possibility of manipulating cells based on their intrinsic dielectric properties and the ease of integrating into microfluidics format, DEP has been widely applied as a label-free cell manipulation and separation method. Thorough reviews on the current state and potential of DEP are available [39,40,41] and progress in integrated microfluidic DEP devices for life science applications, in general, have also been previously reviewed [42,43,44]. Other review articles consider the potential of DEP for next-generation cell sorting [45]. A general overview of different microfluidic separation techniques applicable to microorganisms is provided in [46], especially focusing on bacteria and yeast cells. Here, we review DEP applications for microalgae research and development, including a critical analysis of the advantages and disadvantages of the various DEP microfluidic configurations.

## 3. Overview of DEP Microfluidic Systems for Microalgae Research

Despite a significant amount of work in applying DEP for various microfluidic cell manipulation and separation applications, especially for mammalian cells and bacterial cells, relatively little work has been published in applying the technology for microalgae research [47]. Early works started by Pohl et al. [48,49] involved not only describing the basic DEP formula that governs cell movement under the DEP force but also towards attempting continuous separation of *Chlorella vulgaris* and understanding the dependency of cell movement on media salinity. Some of the earlier works focused on dielectric spectra analysis of cells [50,51,52], as well as basic system development and accompanying electrode designs [53,54].

One of the most interesting and unique characteristics of microalgae compared to other cells that makes them interesting for DEP-based cell manipulation, is that they can accumulate large amounts of intracellular lipid (Figure 3) [55]. As lipid has rather different dielectric properties than typical cytosol, the DEP force generated can be vastly different, making it ideal for applying DEP-based separation and characterization techniques. Additionally, the size of microalgae can also be an indicator of their characteristics or changes in physiology, which offers another excellent opportunity for DEP-based separation. For this review, DEP microfluidic devices for microalgae applications will be critically analyzed from two different aspects; first, based on the microfluidic structures being utilized, and second, based on the applications of such DEP-based microfluidic devices.

A summary of the analyzed publications is listed in Table 1. The table contains information about the microalgae species utilized, device design, and application areas. Two sets of information are provided in detail: the first one focuses on the device and DEP electrode configuration, important aspects when developing DEP-based cell manipulation, separation, and characterization devices. The second one provides details of experimental parameters such as flow velocity, voltages and frequencies applied, cell parameters such as cell size and dielectric properties. Taken together, these two sets of information can serve as a quick lookup table that will be useful in designing specific DEP devices for desired applications of interest. Notation and units are given in Table 2.

From the information summarized in Table 1, several assessments of the current status of the field can be made. First, most of the investigated microalgae belong to the group of green microalgae, although there are some belonging to diatoms (Figure 4). Additionally, most published work has focused on static experiments ($\dot{V}=0$), despite the advantages of continuous-flow operations, such as higher throughput. This shows the challenge of successfully combining dielectrophoretic cell manipulation and separation systems with continuous-flow microfluidic setups. Lastly, in most cases, prior to dielectrophoretic cell analysis or cell manipulation, regular culture media are diluted or exchanged. This can be done by pressure-driven membrane operations, which also show potential in recovering functional molecules during downstream processing [74,75,76]. The decrease in media conductivity results in higher contrast in polarizability $\left(\left|\varepsilon_{p}^{*}-\varepsilon_{m}^{*}\right|\right)$ and thus a larger DEP force. This shows the challenges and limitations of DEP-based cell handling, as in situ applications in normal culture media may become challenging.

## 4. DEP Microfluidic Devices Categorized Based on Working Principles of Devices

There are broadly two different device categories in how DEP-based microalgal cell manipulations are conducted. The first device category traps desired target cells onto the DEP electrodes from cells flowing through a channel using a positive DEP force, essentially functioning as a filtration device that targets specific cells based on their dielectric properties. The second category deflects the cells flowing in a microfluidic channel through either a positive DEP force or a negative DEP force, where the applied force causes the cell trajectories to change, resulting in separation of microalgae cells based on their dielectric properties.

### 4.1. Trapping Designs Using pDEP Force

In these designs, electrodes positioned inside a microfluidic channel applies a pDEP force to cells passing through and traps target cells onto the electrodes. Three different electrodes designs are most commonly utilized.

### 4.2. Planar Parallel Surface Electrodes for Cell Trapping

The first device category is those that trap desired target cells onto the DEP electrodes from cells flowing through a channel. Two DEP electrode structures most commonly used are the planar parallel electrode structure (Figure 5a) and the planar interdigitated electrode structure (Figure 5b). The planar electrode structure is configured to have a flow cell with two parallel electrodes at both sides of the microfluidic channel. Here, the DEP force is applied between the electrodes, thus perpendicular to the flow direction. Suscillon et al. utilized such a structure to trap *Chlamydomonas reinhardtii* cells to the DEP electrodes at an electric field of 20 V mm^−1^ and frequency of 1 kHz [60]. Here the electrode material was gold, which is most commonly used due to its chemical inertness and stability in the solution phase.

Siebman et al. utilized two pairs of wire electrodes positioned at a 180° angle to create a high electric field zone in the middle, into which cells could be trapped [56,61,62]. Here, the electrode was a pair of stainless-steel needles. In these studies, green microalga *Chlamydomonas reinhardtii*, cyanobacterium *Synechocystis sp.*, and diatom *Cyclotella meneghiniana* were utilized, and frequencies from 0.1 to 500 kHz were applied to characterize the cells and their movement. Although easy to assemble as no microfabrication is involved (standard needles were used), it will be difficult to fabricate such devices repeatedly due to difficulties in assembling these four wire electrodes in a reproducible manner, especially when precise cell manipulation is required.

Bahi et al. utilized an interdigitated (IDT) electrode design, where arrays of finger-shaped electrode structures facing each other were utilized [68]. Here the electrode material was platinum. This IDT electrode design is one of the most classical DEP electrode designs used for trapping [77], as the many interdigitated finger structure can generate arrays of high electric field regions, to which cells can be attracted through a positive DEP force (Figure 5b). As a large area can be covered with such an electrode structure, this is a very efficient way of trapping cells. In this application, marine microalga *Karenia brevis* was utilized.

Wang et al. also operated with such an IDT structure [73] in a microfluidic flow cell to trap *Chlorella vulgaris* cells at a voltage of 10 V_pp_ in a frequency range of 0.1 to 5 MHz. Here the electrode material was gold.

Another planar electrode design shown by Kumar et al. had an array of parallel electrodes (modified IDT design) placed on both sides of a flow channel [66], as shown in Figure 5c. Here, cells flowing through the middle part could be attracted to both sides of the microfluidic channel under the influence of dielectric force, trapping target cells to the side of the main flow stream. Since the microfluidic channel widened to both sides of the flow channel, the flow is weak on both sides, thus trapping the cells was easier than having a straight microchannel design. The electrode material was gold. This design was used to investigate the trapping of green microalga *Coscinodiscus wailesii* cells at a voltage range of 1 to 10 V and a frequency range of 1 kHz to 100 MHz.

The advantage of most of these designs is the extremely simple electrode structure and ease of microfabrication since they often require only an electrode design and fabrication process on a printed circuit board (PCB) or a single glass substrate. Additionally, compared to DEP-based cell separation (to be described in the next section), these designs can accumulate and concentrate cells rather than just separate cells, which is useful if concentrating cells from a solution is the main purpose.

However, the major disadvantages are that due to the relatively large distance between the electrodes, the trapping force is relatively weak, limiting the flow rate that can be utilized. IDT electrode design overcomes some of this limitation. However, the electric field and also the dielectrophoretic force, decreases with distance from the electrode. Since in all of these designs the electrodes are on the bottom surface of the microfluidic channel, only cells flowing close to the bottom of a channel can be trapped easily. Cells that are flowing close to the channel ceiling experience a much lower DEP force. Therefore, a relatively shallow microfluidic channel must be used for high trapping efficiency, along with a relatively slow flow rate, thus significantly limiting the overall throughput that can be achieved. A shallow microfluidic channel can also be easily clogged by clumps of cells, posing another practical challenge.

### 4.3. Micropost Electrode for Cell Trapping

Gallo-Villanjueva et al. described a DEP trapping design based on arrays of micro-post-electrodes that can create an asymmetric electric field [69], hence creating zones of high electric field gradients which cells can be attracted to (Figure 5d). Here, the arrays of 20 µm tall posts are made of glass by etching a glass substrate through standard wet etching (20 µm channel depth). The electric field is applied along the flow direction of the microfluidic channel. Thus, the array of insulating posts creates an asymmetric electric field, consequently creating DEP force. This design was utilized to trap the cells of *Selenastrum capricornutum* at an electric field of 250 V mm^−1^ DC potential. A relatively high voltage had to be applied as it is introduced via the in- and outlet of the flow channel.

### 4.4. Sharp-Tip Electrode Design for Cell Trapping

This design, where a sharp electrode is used to create a high electric field at the tip of the electrode is one of the oldest designs and methods to create an asymmetric electric field for DEP experiments [78]. Here, a simple needle-like sharp electrode can be inserted into a solution, and when an electric field is applied, cells can be attracted to this electrode. This can be transferred easily into a microfabricated version, as has been shown by Michael et al. (2014). In this study, the planar sharp-tip electrode made of gold was utilized to trap *Chlamydomonas reinhardtii* cells using high-frequency DEP (>20 MHz). Due to the simplicity, this design is ideal for quickly characterizing the DEP responses of cells, but beyond that, it cannot really be applied for any cell trapping or cell separation applications due to its low throughput nature.

### 4.5. Flow-Through Deflection Structures for Cell Separation

In these designs, electrodes integrated into a microfluidic channel either apply a pDEP or nDEP force depending on frequency. In the overwhelming majority of studies, the microfluidic flow exhibit low Reynold’s numbers that can safely be characterized as laminar. The electrodes then deflect the target cells to one or the other side of the microfluidic channel, resulting in a lateral shift in cell positions inside the flow, thus separating target cells from the initial flow path.

### 4.6. Sharp-Tip Electrode for Cell Separation

As discussed in the previous section, a sharp-tip electrode is one of the earliest forms of electrodes that have been used to excite the inhomogeneous electric field needed to generate the DEP force. Song et al. designed a microfluidic channel structure that has a region where the exterior channel shape resembles a needle [57]. Since the voltage was applied through the channel, this created a large inhomogeneous electric field around the channel tip region. This is similar to having a sharp-tip electrode near the flow channel, without the need to having to create such an electrode. This design was used to deflect *Chlorella vulgaris*, *Pseudokircheriella subcapitata*, and *Dunaliella salina* cells at an applied DC voltage of several hundred volts.

The work from Wang et al. (2018) showed a microfluidic channel where on one side was an array of sharp electrodes and on the other side, a flat planar electrode was positioned. By applying a voltage between the electrodes, this design created an array of inhomogeneous electric field regions on one side of the microfluidic channel. The microfluidic device was composed of a multi-layer polydimethylsiloxane (PDMS) channel with gold electrodes patterned on the glass slide. The electrodes were insulated from the substrate via an underlying silicon nitride layer. Cells that experience a pDEP force (in this case wild type *Chlorella vulgaris*) remained attracted towards the bottom electrode and came out of the lower outlet channel. However, cells that experience a nDEP force (in this case *Chlorella vulgaris* with higher Sr biomineral competence cells) were repelled away from the electrode and exited the upper outlet channel. Based on these two different characteristics, this system was successfully utilized to separate microalgae that show higher radionuclide bio-decontamination activity [79].

### 4.7. Angled Electrodes for Cell Separation

The second category of designs are those where the electrodes are inclined at an angle to the flow direction. Deng et al. (2014) used a device where a pair of top-bottom electrodes were positioned at an angle to the flow direction, where three such electrode pairs were positioned in a zig-zag position along the microchannel. The electrodes were made from gold, and the top/bottom substrates were aligned and bonded together to form the top-bottom electrode structure. The first two pairs were utilized to first align all cells along the electrodes by applying a voltage at 5 MHz to apply strong nDEP to all cells. The third electrode was then utilized to separate cells based on their intracellular lipid content by increasing the frequency to 10 MHz. Here, microalgae with 24% lipid content passed through the electrodes, while those with 35% lipid content were repelled by the nDEP force. Thus, in this device, the frequency-dependent response of cells to experience either a pDEP force or nDEP force was utilized for selective cell manipulation [80].

The most basic device design for the separation of microalgae would encompass a microfluidic channel at the walls of which electrodes are integrated (Figure 5e). The latter would be subjected to oscillating electrical fields to act on particles as carried by the streaming fluid. Figure 2 displays the scheme of a 50 µm high channel with so-called deflector electrodes on the top and at the bottom that was used in a finite-element model to simulate the DEP effect. The DEP force was calculated for *Chlamydomonas reinhardtii* [63] with geometrical and electrical parameters given in the literature [64]. For a flow velocity of v<1.9 µL/min, a frequency of *f* = 1.85 MHz and a voltage of 12 V_pp_ a separation effect could be shown causing the high-lipid fraction (blue) to enter the upper channel and the low-lipid fraction (red) to enter the lower channel.

Hadady et al. (2016) also utilized arrays of angled electrodes to separate microalgal cells based on their properties. Here, arrays of interdigitated angled gold electrodes (45° angle, 50 µm width, 50 µm gap) positioned at the bottom of the flow channel created a sideway DEP force. Consequently, *Chlamydomonas reinhardtii* cells with high lipid content were minimally influenced by the DEP force and went straight out through the lower outlet channel. However, *Chlamydomonas reinhardtii* cells with low lipid content were influenced by the DEP force the most, and, therefore, moved laterally to the upper side of the microfluidic channel, and thus went out through the upper outlet channel. Here, a fairly high frequency of 40 MHz and 60 MHz were utilized. A flow rate of 3 μL/min was utilized throughout this experiment [59].

Overall, the angled DEP electrodes that attract or deflect cells based on their intrinsic properties, especially depending on their intracellular lipid content, provides not only high accuracy in such separation but also significantly higher throughput compared to static capture-type DEP designs. Therefore, these deflector-type devices are probably the most promising DEP microfluidic device design for microalgae manipulation and separation.

## 5. DEP Microfluidic Devices Categorized Based on Their Applications

The various DEP-based microfluidic designs and devices were utilized for the identification of microalgal species, analysis of various properties and responses of microalgae, separation based on their size and intracellular lipid content, as well as for screening applications.

### 5.1. Cell Trapping and Concentration

Suscillon et al. and Siebman et al. from the same group simply demonstrated that microalgal cells can be trapped using DEP [56,60]. In the first study [60], *Chlamydomonas reinhardtii* was trapped between two electrodes under diverse media conditions, voltages, and frequencies, and showed “chaining” behavior. In the second paper [56], this study was expanded to two other microalgal species, *Synechocystis sp.* and *Cyclotella meneghiniana*. However, beyond the observation of this phenomenon, no further specifics were discussed on any specific technical applications where such phenomena may be utilized.

Bahi et al. showed a DEP device to trap marine microalga *Karenia brevis* for lysis and downstream RNA extraction and amplification purposes [68]. The main benefit of the DEP-based microfluidic device was the capability of creating a high concentration of cells, which allowed RNA extraction and purification much easier compared to using low concentration cell samples.

Siebman et al. studied how *Chlamydomonas reinhardtii* cells are impacted by environmental contaminants [61,62]. When cells were exposed to contaminants such as mercury, methylmercury, copper, copper oxide nanoparticles, and the herbicide diuron, reactive oxygen species production and oxidative stress increased in *Chlamydomonas*, which were measured by detecting the changes in chlorophyll autofluorescence. In this application, the DEP trapping structures were utilized to hold onto cells while the cells are exposed to the environmental contaminants and while taking fluorescent images, allowing easy microscopy.

### 5.2. Cell Separation Based on Intracellular Lipid Content

Since lipid production by microalgal cells is one of the major reasons why microalgal research is of high interest, several publications have described the use of DEP to separate microalgal cells based on their intracellular lipid content.

Deng et al. presented two such studies [58,80]. In the first work, *Chlorella vulgaris* cells with different lipid content were successfully separated using a DEP device having arrays of parallel electrodes. Here, the DEP microfluidic system demonstrated that *Chlorella* with 11 wt% and 45 wt% lipid content showed very different dielectric properties that could be successfully separated. This DEP separation was performed at a relatively high frequency of 20 MHz. In the group’s second paper, *Chlorella* cells with lipid content of 13% and 21% were separated, and cells with lipid content of 24% and 30-35% were successfully separated. In this work, slightly lower frequencies of 7 MHz and 10 MHz were utilized for cell separation.

In another work, Michael et al. used a relatively high frequency (over 20 MHz) to show that the upper crossover frequency of microalgal cells is reduced with increasing intracellular lipid [81]. This phenomenon was successfully measured on *Chlamydomonas reinhardtii* cells using a sharp-tip electrode device. Later, the same group utilized this phenomenon and applied it to a flow-through DEP device [59], where 74% of the high-lipid population and 75% of the low-lipid population could be successfully separated. In this particular device, again a relatively high frequency of 50 MHz was utilized at a voltage of 30 V_pp_.

### 5.3. Cell Separation Based on Their Sizes or From Other Particles

Song et al. presented a device that they used to separate marine microalgal species known to have different volumes from 5 µm diameter polystyrene (PS) particles [57]. The justification for this work was for analyzing bioparticles for water quality monitoring. These were *Chlorella vulgaris*, *Pseudokircheriella subcapitata*, and *Dunaliella salina*. *P. subcapitata* and *D. Salina* had a similar size to the 5 µm diameter PS particles, while *C. vulgaris* was in the range of 2–4 µm in diameter. Here, *C. vulgaris* and *P. subcapitata* were successfully separated into two streams using their DEP microfluidic device. Additionally, the DEP separation of *P. subcapitata* from 5 µm diameter PS beads and the DEP separation of *D. Salina* from 5 µm PS beads were also demonstrated. These separations were all based on size differences between the cells and the PS beads.

Wang et al. also demonstrated the separation of *Platymonas* and *Closterium* from microplastic [73]. Here, the application was to select microalgae from ballast water in a ship, as biological contamination of ballast water is a global problem as well as highly regulated, where ballast water must be inactivated for any biological materials before docking into a port. Thus, rapid detection of ballast water for any microalgae is needed and has the potential to overcome the challenges of using filtration systems and fluorescence measurement. Here, the group demonstrated that both *Platymonas* and *Closterium* can be successfully separated from Polystyrene particles (size not mentioned in the manuscript), using a voltage range of 5–15 V at a frequency of several tens of MHz. The system was also tested under various flow rates, with the maximum being 0.03 mL/min. Overall, separation efficiencies of around 90% were achieved.

### 5.4. Microalgae Analysis

Kumar et al. utilized a DEP device to measure the dielectric properties of a green alga *Coscinodiscus wailesii* [66]. Here, the lateral displacement, dielectrophoretic force, and translational dielectrophoretic velocity were measured. The authors concluded that thoroughly measuring the dielectric properties of a given cell provides the future capability in label-free manipulation of diatoms and for rapid screening for environmental effects on the dielectric properties of algal cells (but not yet conducted in their paper).

Gallo-Villanueva et al. utilized a DEP device to measure the viability of *Selenastrum capricornutum* [69]. The experimental result showed that live cells exhibit a stronger DEP response compared to dead cells. This allowed rapid label-free sensing of the viability of the microalgal cells. The authors presented that within a relatively short period of time (35 s), enrichment of about 10 times could be achieved for each cell population.

### 5.5. Strain Selection Through Screening

Selecting microalgal strains with specific phenotypes is of high interest in many microalgae screening applications. Many such applications have been on selecting and obtaining high-lipid-producing strains, as microalgae-based lipid production is an important area towards renewable bioenergy applications [19,65]. However, there are other examples of high-throughput screening than just lipid production.

An example of such an application is identifying strains that show high efficiency in decontaminating hazardous waste, such as radionuclide. Wang et al. used a microfluidic device to first test the capability of DEP for their capability in separating *Chlorella vulgaris KMMCC9*, a strain known for its high capacity of removing strontium (Sr), from *Chlorella vulgaris KCTC AG10002* strain that has only a weak capacity of removing strontium. This separation was successfully conducted using the developed DEP device [79]. Following this success, sub-populations of *Chlorella vulgaris KMMCC9* strains with higher Sr biomineral competence were successfully separated, and the selected strain’s capability confirmed in large-scale cultivation experiments.

## 6. Discussion

From the analysis of DEP-based microfluidic platforms for microalgae research, it can be concluded that DEP in microalgae research is promising but still not a mainstream technology. The latter fact is caused by several reasons. First, the manufacturing complexity of microfluidic DEP devices is still not standard technology, especially to non-experts in microfabrication. Several works feature the need for highly specialized microfabrication equipment or non-standardized/non-comparable setups. Some of these reasons can be contributed to the fact that microfluidics devices have no standardized guidance, which has been a fundamental limitation for microfluidic systems in general. This, in general, has hindered many excellent microfluidic systems from being adopted by the broader life science community. The functional part of a microfluidic DEP setup, in general, has in most cases not exceeded technology readiness level 4, which stands for a principle demonstration in a laboratory setup. Despite these limitations, microfluidic systems have been proliferating thanks to their powerful and unique capabilities, as well as better availability of such devices through commercial vendors that provide not only certain pre-designed microfluidic chips, but also custom microfluidic chips designed by researchers and fabricated through foundry services.

Second, and closely related to the first reason, is the rather high operational complexity of microfluidic systems. In addition to the external instruments required to drive the microfluidic chips that may not be readily available in many life science laboratories, methods and protocols are developed within individual research groups and rarely adapted or proven by others. However, this is also becoming less of a bottleneck as many external instruments that drive microfluidic devices (e.g., syringe pumps) are becoming significantly less expensive, more standard operation procedures are becoming available through the large number of microfluidics-based papers that are being published, as well as detailed protocol-type papers being published (e.g., through the Journal of Visualized Experiments), and the large number of microfluidics-based papers being published that are specifically focused on simple device structures or ease of operation for non-microfluidic experts in mind.

Third, more specifically to DEP microfluidic devices for microalgae applications, various physical limitation needs to be considered. The local electric field gradient is crucial for the DEP force. However, high voltage results in the heating of cells that can negatively impact the cells or bubble generation on electrodes that make the DEP microfluidic device not function properly. Additionally, since the DEP force relies on differences in dielectric properties of target cells versus that of the surrounding media, most applications so far have utilized low-conductivity media to increase the DEP force. This makes it difficult to apply DEP microfluidic systems for various applications where in situ measurements are desired or even required. As electrical field responses of biological cells are dependent on the applied frequency, the capability of accurately measuring DEP responses over large frequency ranges are gaining importance for applying DEP microfluidic to recent microalgae research.

Fourth, biological parameters also need consideration. Unlike many mammalian cells, where DEP microfluidic devices have been extensively developed and utilized, many microorganisms, including types of microalgae, have non-spherical shapes. This adds complexity to not only their DEP responses but also difficulties in accurately predicting and simulating the movement of such cells under the influence of the DEP force. Various sub-cellular structures, such as intracellular lipid droplets within these microalgae or external flagella, can also pose significant challenges. Cells that move actively due to flagella could especially pose challenges for DEP-based cell manipulation and separation.

Finally, another category of biological parameters to consider is the fact that microalgal cell sizes can vary severely depending on their physiological state. As can be seen in Equation (1), both the cell size, as well as the dielectric properties of the cell, influence the degree of the DEP force. Some microalgae, such as *Chlamydomonas reinhardtii*, show a rather constant cell size regardless of their physiological state e.g., exponential growth phase vs. stationary phase, lipid induction state vs. growth phase. However, some microalgae cells such as *Crypthecodinium cohnii* can exhibit significantly different cell sizes depending on their growth phase or levels of intracellular lipid content. Thus, having to decouple cell size-dependent DEP effect from other phenotypes of interest such as intracellular lipid-dependent DEP effect becomes important. As most cells have heterogeneous subpopulations, such heterogeneity further adds to the challenges.

Despite these challenges, the unique capability of DEP-based microfluidic systems in enabling label-free manipulation of cells depending on their intrinsic properties remains an extremely attractive method for microalgal research. Thus, we expect to see significantly more development in DEP microfluidic systems specifically targeting microalgae in the near future.

## 7. Conclusions

Microalgae-based bioproduction of high-value biomolecules, including those for transportation fuel, is a promising avenue towards achieving a higher degree of bioeconomy, however, it is currently struggling with profitability and, therefore, commercialization. Conventional methods utilized throughout the microalgae-based bioproduct development pipeline have failed so far to deliver economically viable products in most cases [82]. To achieve the necessary yields and efficiencies, new or improved microalgal strains are needed. Many published works on DEP-based microfluidic devices applied to microalgae have shown high potentials for applications in the fields of strain development, process analysis, and characterization. Here, we have provided a comprehensive review and analysis of DEP-based microfluidic devices utilized for microalgae research, with an in-depth analysis of the various device categories and application areas of such works. We have also provided a critical analysis of the advantages and disadvantages of each method, with a concluding remark on the perspective and future of DEP-based microfluidic devices for microalgae research. 

## Figures and Tables

**Figure 1 microorganisms-08-00540-f001:**
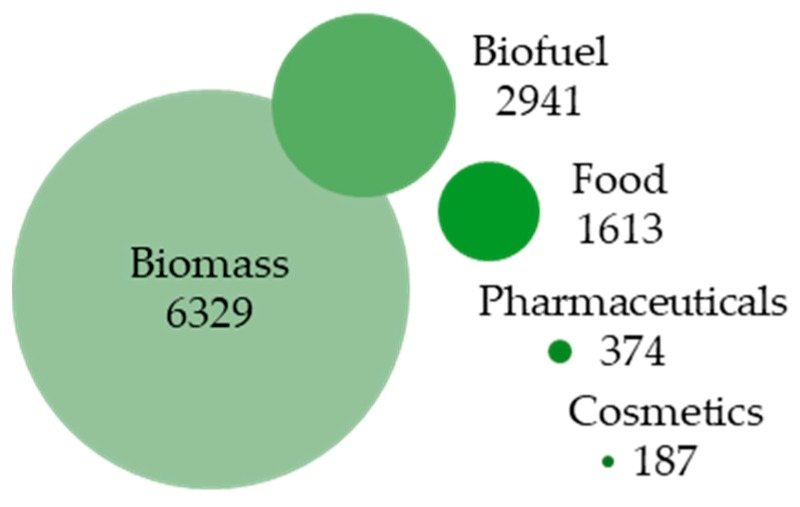
Research areas of publications associated with the term “microalgae“. Bubbles show the number of results of combined searches “microalgae + indicated term” based on publications of the past five years [Source: webofknowledge.com].

**Figure 2 microorganisms-08-00540-f002:**
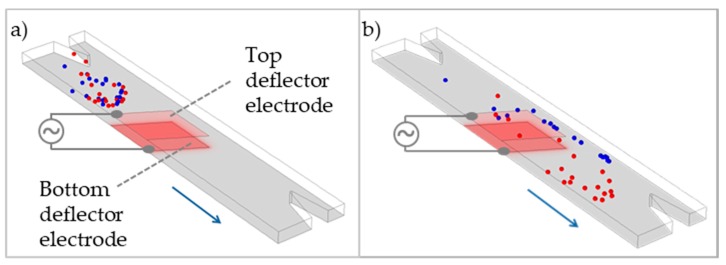
Simulation of dielectrophoresis (DEP) cell separation in a microfluidic flow. (**a**) before and (**b**) after microalgae *Chlamydomonas reinhardtii* have passed a deflector electrode configuration (top and bottom electrode).

**Figure 3 microorganisms-08-00540-f003:**
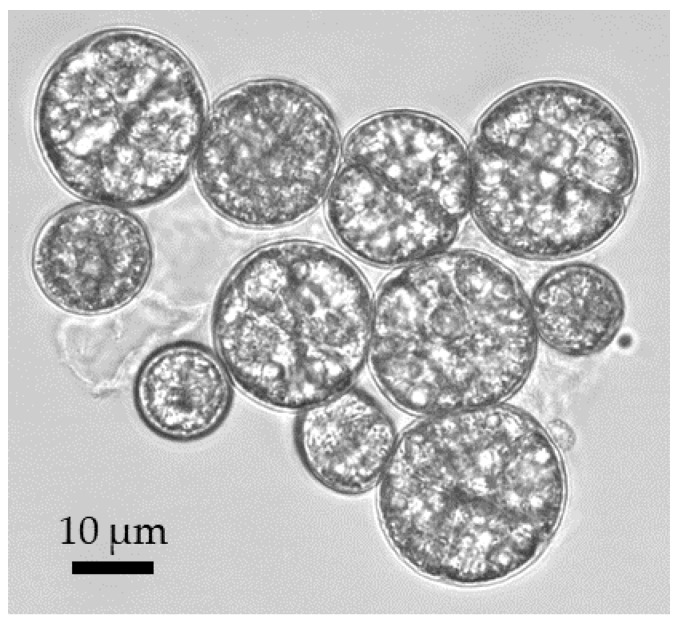
Light microscopic images of the microalga *Crypthecodinium cohnii*.

**Figure 4 microorganisms-08-00540-f004:**
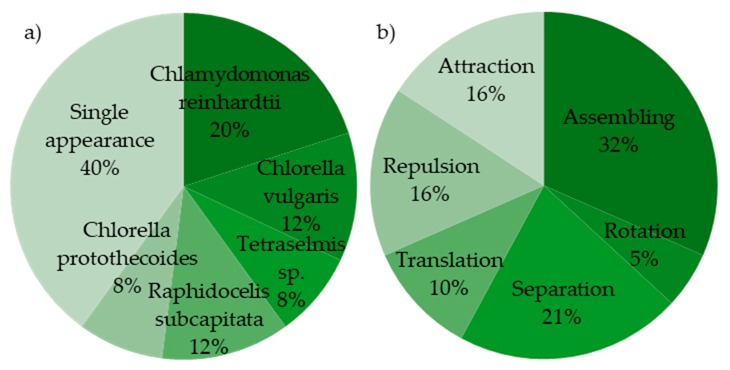
Overview of microalgae DEP research based on 16 publications (Table 1). (**a**) Used species of algae, (**b**) Application types of DEP in microalgae research.

**Figure 5 microorganisms-08-00540-f005:**
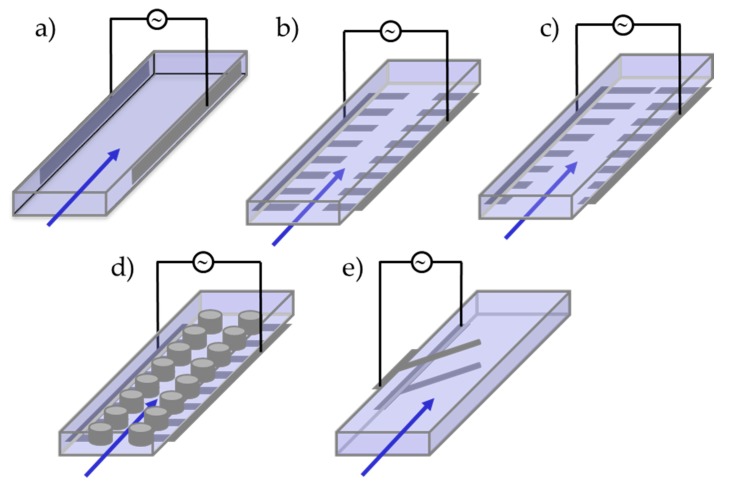
Schematics of channel-electrode configurations used in DEP studies on microalgae. Electrode configurations are (**a**) planar parallel, (**b**) planar interdigitated, (**c**) modified planar interdigitated, (**d**) micro-post and (**e**) angled.

**Table 1 microorganisms-08-00540-t001:** Key publications reviewed in this article, with a summary of key features and applications.

Algal Species	Taxonomy and No. Flagella	Application Type and Description		Device Structure	Electric Field, Flow Rate, and Cell Concentration	Dielectric Properties of the Medium	Dielectric Properties of Microalgae	Ref.
(a) *Chlamydomonas reinhardtii* (b) *Synechocystis sp.* (c) *Cyclotella meneghiniana*	(a) green alga, 2 (b) cyanobac., 0 (c) diatom, 0	ass. rot.	measuring effects of AC field intensity, frequency and duration on chaining efficiency and chain lengths	chamber $\left(d_{\mathrm{ch}}=350\right)$, coplanar electrodes $\left(\mathrm{Au},{\;d}_{\mathrm{eg}}=2000\right)$	$\begin{array}{l} E=15\ldots 25\\ f=10^{-4}\ldots 0.5\\ \dot{V}=0\\ c=5\cdot 10^{6}\ldots 5\cdot 10^{7} \end{array}$	Geneva lake water $\sigma_{m}=320$ $\varepsilon_{m}=80$	$\left(a\right)\;d=13,{\;t}_{w}=500,\;{\;\sigma }_{i}=0.008,\sigma_{w}=50,\;{\;\varepsilon }_{i}=150,\varepsilon_{w}=70$ $(b)\;d=3.98,{\;t}_{w}=130,\;\;\sigma_{i}=0.19,{\;\sigma }_{w}=680,\;{\;\varepsilon }_{w}=60,{\;\varepsilon }_{i}=61$ $(c)\;d=17.72,t_{w}=500,\;\;\sigma_{i}=0.008,{\;\sigma }_{w}=10^{-17}$ $\varepsilon_{i}=150,{\;\varepsilon }_{w}=3.9$	[56]
(a) *Chlorella vulgaris* (b) *Raphidocelis subcapitata* (c) *Dunaliella salina*	(a) green alga, 0 (b) green alga, 0 (c) green alga, 2	sep.	separation by size and species	PDMS channels $\left(d_{\mathrm{cw}}=90\ldots \;300,{\;d}_{\mathrm{ch}}=25\right)$ with overall field gradient	$\begin{array}{l} U<295\\ f=0 \end{array}$	sodium borate buffer solution pH 7.5	(a) d=2…4 (b) d=3.7…6.25 (c) d=3.8…6.0	[57]
*Chlamydomonas reinhardtii* (a) high lipid (b) low lipid	green alga, 2	sep.	high-frequency DEP in continuous-flow cell screening device for separation based on lipid content	$\mathrm{PDMS}\;\mathrm{channel}\;(d_{\mathrm{ch}}=20,{\;d}_{\mathrm{cw}}=1000),$ 4 interdigitated electrode arrays 10 electrodes each $\left(\mathrm{Au},\;d_{\mathrm{ew}}=50,{\;d}_{\mathrm{eg}}=50\right)$ by etching	U=30 f=50 $\dot{V}=9$ $c=6.7\cdot 10^{6}$	KCl solution $85{\;g\;L}^{-1}\;\mathrm{Glc}$ 0.1% serum albumin $\sigma_{m}=10.6$ $\varepsilon_{m}=80$	d=10… 15 $\left(a\right)\;\sigma_{i}=0.095$ $\left(b\right)\;\sigma_{i}=0.2267$ $\varepsilon_{i}=50,$ $\varepsilon_{\mathrm{mem}}=8$ $\sigma_{\mathrm{mem}}=2\cdot 10^{-6}$	[58]
*Chlamydomonas reinhardtii*	green alga, 2	ass.	high-frequency DEP to determine upper crossover frequency of cells with varying lipid content	glass slide with needle patterned electrodes (Au)	U=30 f=10…110 $\dot{V}=0$	KCl solution; $85{\;g\;L}^{-1}\;\mathrm{Glc};$ $\sigma_{m}=64$ $\varepsilon_{m}=80$	$d=12,{\;t}_{\mathrm{mem}}=7\;\varepsilon_{i}=50,{\;\varepsilon }_{\mathrm{mem}}=8,\;\;\sigma_{i}=0.5,\;\sigma_{\mathrm{mem}}=0.02$	[59]
*Chlamydomonas reinhardtii*	green alga, 2	ass.	characterization of effects of freshwater composition on the DEP response	$\mathrm{chamber}\;\left(d_{\mathrm{ch}}=250\right)$, coplanar electrodes $\left(\mathrm{Au},{\;d}_{\mathrm{eg}}=2000\right)$ by vapor deposition	E=20 f=0.001 $\dot{V}=0$ $n=10^{6}$	fresh water; $\sigma_{m}=32\ldots \;56$		[60]
*Chlamydomonas reinhardtii*	green alga, 2	ass.	rapid tool for capture and screen with fluorescence for the effect of contaminants	$\mathrm{chamber}\;\left(\mathrm{silicone},d_{\mathrm{ch}}=2000\right),$ $\mathrm{four}\;\mathrm{orthogonally}\;\mathrm{needle}\;\mathrm{electrodes}\;\left(\mathrm{stainless}\;\mathrm{steel},{\;d}_{\mathrm{eg}}=5000\right)$	E=10 f=0.0001 $\dot{V}=0$ $n=5\cdot 10^{6}$	water; 0.0001 M MOPS +various contaminants		[61,62]
*Chlorella vulgaris*	green alga, 0	sep.	studies on solution conductivity and lipid content, microfluidic chip to sort the microalgae with different lipid contents	$\mathrm{channel}\;\left(\mathrm{JSR}\;\mathrm{THB}151N,{\;d}_{\mathrm{ch}}=15\right)$ by spin coating, symmetrical deflector electrodes $\left(\mathrm{Ti}\;0.2\;\mathrm{Au},{\;d}_{\mathrm{eh}}=0.03\;\right)$ by vapor deposition and photolithography	U=10 f=7, 10, 20 $\dot{V}=250$	${\mathrm{KH}}_{2}{\mathrm{PO}}_{4}$ buffer solution; $\sigma_{m}=290$ $\varepsilon_{m}=80$	d=5.2 $d_{\mathrm{lip}}=1.23\ldots 2.05$ $\sigma_{w}=10^{-8},\;\sigma_{i}=0.5$ $\sigma_{\mathrm{lip}}=0.0001$ $\varepsilon_{w}=5,{\;\varepsilon }_{i}=60$ $\varepsilon_{\mathrm{lip}}=3$ $t_{w}=100$	[63,64]
*Chlorella vulgaris*	green alga, 0	att. rep.	screening for highest radionuclide bio-decontamination by n- and p-DEP	$\mathrm{PDMS}\;\mathrm{Chamber},\;\mathrm{electrodes}\;\left(\mathrm{Au},{\;d}_{\mathrm{eh}}=0.3,{\;d}_{\mathrm{ew}}=30\right)$ on glass by lithography	U=10 f=0.1…5 $c=4.8\cdot 10^{6}$	$3{\;\mathrm{mM}\;\mathrm{NaHCO}}_{3}$ $\sigma_{m}=335$		[65]
*Coscinodiscus wailesii*	diatom, 0	att. rep.	2D dielectrophoretic signature	PDMS microfluidic well, interdigitated electrode pattern	U=1…10 f=0.001…100 $\dot{V}=0$	f/2 culture medium $\sigma_{m}=47$ $\varepsilon_{m}=79$	d=20,75… 80 $\sigma_{i}=0.06,{\;\sigma }_{\mathrm{mem}}=0.03$ $\varepsilon_{i}=48,{\;\varepsilon }_{\mathrm{mem}}=20$ $t_{\mathrm{mem}}=9$	[66]
*Eremosphaera viridis*	green alga, 0	att. rep.	tool for spatial manipulation	commercially available single electrode, etched elgiloy tip with porous metal-oxide coating	U=1…5 f=0.05…10 $\dot{V}=0$	low calcium Dickinson medium		[67]
*Karenia brevis*	dinoflagellate, 2	ass.	dielectrophoretic concentration	$\mathrm{glass}\;\mathrm{slide},\;3x4\mathrm{mm}\;\mathrm{array}\;\mathrm{of}\;\mathrm{castellated}\;\mathrm{interdigitated}\;\mathrm{electrodes}\;\left(\mathrm{Pt},{\;d}_{\mathrm{eh}}=0.2,{\;d}_{\mathrm{ew}}=20,{\;d}_{\mathrm{eg}}=20\right)$	U=1 f=0.2 $\dot{V}=0$ $n=3\cdot 10^{5}$	280mM mannose; 0.5% Tween $\sigma_{m}=10.5$		[68]
*Raphidocelis subcapitata*	green alga, 0	sep.	concentrate and separate live and dead cells	$\mathrm{glass}\;\mathrm{chambers}\;\left(d_{\mathrm{cw}}=2000,{\;d}_{\mathrm{ch}}=20\right)$with cylinders (470… 520 µm) by wet etching, overall field gradient	E=10…25 f=0 $n=1.7\cdot 10^{7}$	(i) bidistilled water (ii) 1mM KH2PO4 $\sigma_{m}=0.225,\;18.7$		[69]
*Tetraselmis sp.*	green alga, 4	trans.	twDEP used to estimate the dielectric properties	glass slide, octa-pairs interdigitated electrode $\left(\mathrm{Au},\;\;d_{\mathrm{ew}}=50,{\;d}_{\mathrm{eg}}=50,{\;d}_{\mathrm{eh}}=0.5\right)$ by photolithography and wet-etching	U=1.5…14 f=0.005…4 $\dot{V}=2.4$ $c=10^{6}$	0.5 M sorbitol solution +0.1 M KCl $\sigma_{m}=3\ldots 370$		[70]
*Tetraselmis sp.* (a) control (b) As treated (c) boiled	green alga, 4	trans.	determination of dielectric properties and effects of arsenic	glass slide, octa-pair interdigitated electrodes $(\mathrm{Au},\;\;d_{\mathrm{el}}=200;{\;d}_{\mathrm{ew}}=100,{\;d}_{\mathrm{eh}}=0.2;{\;d}_{\mathrm{eg}1}=100,\;300$)	U=2…10 f=0.015…0.5 $\dot{V}=0$ $c=4\cdot 10^{5}\ldots 9.2\cdot 10^{6}$	$\sigma_{m}=10\ldots 250$ $\varepsilon_{m}=78$	$shperic:d_{a}=20$ $d_{b}=16$ $\left(a\right){\;\sigma }_{i}=0.37$ $\sigma_{\mathrm{mem}}=0.00017$ $\varepsilon_{i}=48,{\;\varepsilon }_{\mathrm{mem}}=8$ $\left(b\right){\;\sigma }_{i}=0.013\ldots 0.03$ $\sigma_{\mathrm{mem}}=0.003\ldots 0.0038$ $\varepsilon_{i}=48,{\;\varepsilon }_{\mathrm{mem}}=10\ldots 32$ $\left(c\right)\;\sigma_{i}=0.06$ $\sigma_{\mathrm{mem}}=0.03$ $\varepsilon_{i}=91,{\;\varepsilon }_{\mathrm{mem}}=20$ $t_{mem}=13$	[71]
heterogeneous population	various	ass.	technique to monitor the concentration of algae in fresh water to avoid mass contaminations	chamber, four electrodes	U<1.65 f<1.2 $\dot{V}=0$		d=15	[72]
(a) *Platymonas sp.* (b) *Closterium sp.*	(a) green alga, 2 (b) green alga, 0	sep.	continuous separation of different microalgae from microplastics by multi-electrode n- and p-DEP	$\mathrm{PDMS}\;\mathrm{chamber}\;\left(d_{\mathrm{cw}}=100\right),\;$ $\mathrm{electrodes}\;\mathrm{on}\;\mathrm{ITO}\;(\mathrm{Ag}-\mathrm{PDMS}\;\mathrm{mixture},{\;d}_{\mathrm{ew}1}=1900,\;$ $d_{\mathrm{ew}2}=100,{\;d}_{\mathrm{eg}}=100)$	f=30 $\dot{V}=0.083$	PBS buffer solution $\sigma_{m}=300$		[73]

**Table 2 microorganisms-08-00540-t002:** Notation and Units.

Term	Meaning	Unit	Term	Meaning	Unit/Value	Term	Meaning	Unit
d	Cell diameter	μm	σm	Suspending medium conductivity	mS·m−1	tw	Cell wall thickness	nm
dew	Electrode width	μm	σmem	Cell membrane conductivity	mS·m−1	tmem	Cell membrane thickness	nm
deh	Electrode height	μm	σi	Cell interior conductivity	S·m−1	c	Cell concentration	ml−1
del	Electrode length	μm	σlip	Cell lipid body conductivity	S·m−1	n	Cell number	-
ded	Electrode diameter	μm	σw	Cell wall conductivity	S·m−1	V˙	Flow rate	μl·s−1
deg	Gap to next electrode	μm	εm	Suspending medium relative permittivity	-	f	frequency	MHz
dcw	Channel/chamber width	μm	εi	Cell interior relative permittivity	-	E	Field strength	V·mm−1
dch	Channel/chamber height	μm	εlip	Cell lipid body relative permittivity	-	U	Voltage	V
dcl	Channel/chamber length	μm	εw	Cell wall relative permittivity				
dcl	Channel/chamber length	μm	ε0	Permittivity of free space	8.854×10−12 F·m−1			
dcd	Channel/chamber diameter	μm

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
