# Peer review of "Separation, Characterization, and Handling of Microalgae by Dielectrophoresis"

_microorganisms, 2020, doi:10.3390/microorganisms8040540_

Round 1

Reviewer 1 Report

The paper entitled "Separation, characterization and handling of microalgae by dielectrophoresisis" provides interesting insights for the readers in the field. This review paper provides a comprehensive review and analysis on DEP-based microfluidic devices used for microalgae research, with an analysis on the various device types and application areas. The authors have provided a critical analysis of the advantages and drawbacks of each method, providing the concluding remarks and future trends DEP-based microfluidic device for microalgae research. 

From my point of view, the article can be accepted for publication after attending some minor comments.

1) Please, double-check the grammar of your paper, there are several typos.

2) The authors should incorporate brief feedback about some pre-treatment unit operations needed before DEP separations, e.g. pressure-driven membrane operations (MF, UF and NF). The authors can use the following suggested reviews for such feedback, please reference them in your revised manuscript:

https://doi.org/10.1016/j.tifs.2019.12.003

DOI: 10.1002/apj.2332

http://dx.doi.org/10.1016/j.foodchem.2016.07.030

Author Response

Dear Reviewer,
thank you very much for your time and effort in reviewing our manuscript.

As suggested we repeatably went through the manuscript for language improvement. Also, thank you for your second point. Sample preparations plays a major role and is credited as one of the main challenges in adaptation of microfluidics. We gladly added the provided reviews at the end of chapter 3.

Reviewer 2 Report

It is an up to date and very well documented review. Congratulations to the authors.

Author Response

Dear Reviewer,
thank you very much for your kind words and your execptional rating.

Regards

Vinzenz Abt

Reviewer 3 Report

This review paper is well edited from basic/fundamental knowledge of DEP and application of DEP for characterization, manipulation and separation of microalgae. The review summarizes various related papers. The review paper provide valuable knowledge/insights and will be useful for microalgal biotechnology and will further develop the microalgal biotechnology.

Some comments:

Fig. 1: How did you create this figure? “Size of bubble correlates to relevance in literature according to the author’s impressions” is unclear and not scientific. The size means the number of literature previously reported? Also “complexity” is unclear. The components are complex or the producing processes are complex?

Fig. 5: It is better to show the 30 review publications.

Author Response

Dear Reviewer,
thank you for your time and effort in reviewing an providing us with valuable feedback.

We went through the document and improved the language. Also, we improved figure 1 and went from our own impression towards a quantitative approach to give insight the various fields of microalgae research. Similarly, we updated figure 5, as it previously included more papers then the final table 1 consisted of.

Thank you again and best regards
Vinzenz Abt